# Lipid Droplet Formation Is Regulated by Ser/Thr Phosphatase PPM1D via Dephosphorylation of Perilipin 1

**DOI:** 10.3390/ijms231912046

**Published:** 2022-10-10

**Authors:** Rui Kamada, Sae Uno, Nozomi Kimura, Fumihiko Yoshimura, Keiji Tanino, Kazuyasu Sakaguchi

**Affiliations:** 1Laboratory of Biological Chemistry, Department of Chemistry, Faculty of Science, Hokkaido University, Sapporo 060-0810, Japan; 2School of Pharmaceutical Sciences, University of Shizuoka, Shizuoka 422-8526, Japan; 3Laboratory of Organic Chemistry II, Department of Chemistry, Faculty of Science, Hokkaido University, Sapporo 060-0810, Japan

**Keywords:** phosphatase inhibitor, lipid droplet, dephosphorylation, adipocytes

## Abstract

Hypertrophy and hyperplasia of white adipocytes induce obesity, leading to diseases such as type 2 diabetes and hypertension, and even cancer. Hypertrophy of white adipocytes is attributed to the excessive storage of the energy form of triglycerides in lipid droplets (LDs). LDs are fat storage organelles that maintain whole-body energy homeostasis. It is important to understand the mechanism of LD formation for the development of obesity therapy; however, the regulatory mechanisms of LD size and formation are not fully understood. In this study, we demonstrated that the PPM family phosphatase PPM1D regulates LD formation. PPM1D specific inhibitor, SL-176 significantly decreased LD formation via two different pathways: dependent of and independent of adipocyte-differentiation processes. In the mature white adipocytes after differentiation, LD formation was found to be controlled by PPM1D via dephosphorylation of Ser511 of perilipin 1. We found that inhibition of PPM1D in mature white adipocytes significantly reduced the size of the LDs via dephosphorylation of Ser511 of perilipin 1 but did not change the lipolysis sensitivity and the total amount of lipid in cells. Collectively, the results of this study provide evidence that PPM1D plays an important role in LD formation in mature adipocytes.

## 1. Introduction

One in five people in many developed countries suffer from obesity [1]. Obesity is a global challenge because it leads to chronic diseases, such as cardiovascular disease and diabetes, and is associated with the occurrence of cancer. Obesity is caused by hypertrophy of adipocytes that is induced by an increase in lipid accumulation and hyperplasia, which occur due to the increased proliferation of white adipocytes. Excessive formation of lipid droplets (LDs) in white adipocytes leads to adipocyte hypertrophy. Thus, it is essential to understand the regulatory mechanism of LD formation, which can aid in the development of obesity treatment [2].

White adipocytes have large LDs, which are fat storage organelles that play an important role in energy metabolism in cells. The formation of LDs is strictly regulated, especially in adipocytes. In mature white adipocytes, LDs continuously form and degrade in response to energy intake and lipolysis stimuli to accumulate and release lipids. Therefore, to understand and control LD formation, it is necessary to reveal the mechanism of LD formation via the adipocyte differentiation-independent pathway in mature white adipocytes. Various cells have LDs, and the size of LDs depends on the cell type. The size of LDs is also related to various pathologies, including cancer, obesity, and diabetes mellitus [3,4]. LD size has been associated with the sensitivity to lipolysis and lipophagy catabolism in hepatocytes [5]. Lipolysis targets large LDs compared to small LDs. In contrast, another study reported that relatively small LDs tend to undergo lipolysis [6]. Thus, it was suggested that the sensitivity of lipolysis of LDs is regulated by other control factors, and not simply determined by its size. However, it is unclear how the size of LDs and lipolysis are regulated in different cell types.

LDs are formed during adipocyte differentiation. When preadipocyte cells are stimulated for differentiation, expression of adipogenic transcription factors, CCAAT/enhancer binding protein b (C/EBPβ), and C/EBPδ is increased, and these transcription factors activate the transcription of C/EBPα and peroxisome proliferator-activated receptor γ (PPARγ) [7,8,9]. C/EBPα and PPARγ have been reported to regulate the expression of each other [10]. These two transcription factors regulate the expression of downstream genes that are required for LD formation, such as the PAT family protein perilipin 1.

In mature white adipocytes, various proteins are localized on the surface of LDs and control fusion, growth, fission, and lipolysis. Perilipin 1, a downstream target of PPARγ, is one of the major regulatory proteins involved in LD formation [11]. Perilipin 1 is localized on the surface of LDs and prevents lipase from binding to LDs for lipolysis. Perilipin 1 is specific to the adipose tissue and testis [12]. It has several phosphorylation sites in its C-terminal region, and phosphorylation at these sites is suggested to regulate specific functions of perilipin 1. Perilipin 1 has been reported to play an important role in lipid storage, and perilipin 1 knockout mice have been reported to exhibit a lean phenotype [13]. The size of LDs was increased by contacting and fusing two LDs of different sizes. It has been reported that perilipin 1 is also involved in LD fusion [14]. The fat-specific protein 27 (Fsp27) is reported to be present on the LD contact surface at the time of LD fusion; perilipin 1 promotes LD fusion by changing the conformation of the Fsp27 dimer. In contrast, perilipin 1 is also involved in lipolysis activation [15]. Perilipin 1 is phosphorylated by PKA, in response to adrenergic receptor-mediated lipolysis stimulation, which leads to activation of lipolysis [16]. Thus, phosphorylation of perilipin 1 is an important factor that regulates LD formation, lipolysis, and LD fusion. Protein phosphatase 1 (PP1) has been reported to be involved in the dephosphorylation of perilipin 1; however, the target site of PP1 is unknown [17]. The involvement of enzymes that dephosphorylate perilipin 1 in LD formation remains unclear.

Ser/Thr phosphatase PPM1D (Wip1) was identified as a tumor suppressor protein p53 inducible protein [18]. PPM1D has been reported to be associated with cellular metabolism. Atherosclerosis was reported to be induced in apolipoprotein E (ApoE)-deficient mice, but arteriosclerosis and fat accumulation in macrophages were suppressed in ApoE and PPM1D double knockout mice [19]. In these PPM1D knockout mice, suppression of weight gain and fat accumulation were observed in high fat diet-fed mice. The expression level of PPM1D protein increases in mice models of obesity, suggesting that PPM1D is involved in lipid metabolism and obesity [20]. We previously reported a strong and specific PPM1D inhibitor: SL-176 [21]. We found that in preadipocyte 3T3-L1 cells, differentiation into white adipocytes in the presence of SL-176 significantly reduced the amount and size of LDs and reduced the mRNA expression of adipocyte differentiation markers [22]. However, the function of PPM1D in mature adipocytes and LD formation remains unclear. Here, we demonstrated that PPM1D inhibition in pre-adipocytes significantly reduced the LD formation by suppressing adipocyte differentiation. Moreover, we found that PPM1D regulates formation of LD by dephosphorylating Ser511, a novel dephosphorylation site of perilipin 1. Our results demonstrated that the PPM1D inhibitor SL-176 reduced LD formation both in pre-adipocytes and mature adipocytes. Together, these results demonstrate that PPM1D phosphatase is a novel target for regulating LD formation in adipocyte.

## 2. Results

### 2.1. PPM1D Inhibition in Early Stage Suppressed Adipocyte Differentiation

We have reported that white adipocyte differentiation is significantly inhibited when the PPM1D inhibitor SL-176 is added during the differentiation of 3T3-L1 cells to inhibit the phosphatase activity of PPM1D [22]. To determine which step among the early, middle, and late stages of adipocyte differentiation PPM1D is involved in, the PPM1D inhibitor was added at different stages of adipocyte differentiation, and the amounts of LDs, which represent the differentiation induction efficiency, was measured by Oil red O staining (Figure 1A–C). Inhibition of PPM1D in the early stage of differentiation, at days 0–3 and days 0–5, strongly reduced the amounts of LDs to 80% and 60% compared with mock-treated cells, respectively. In contrast, inhibition in the late stage of differentiation, on days 5–8, did not affect the amount of LDs.

Next, we performed LD imaging using a fluorescent probe for LD and monodansylpentane (MDH) (Figure 2A,B). Staining with MDH showed that LD size was significantly decreased by SL-176; the average size of LDs in mock-treated cells was 2.60 µm in diameter, and the average size of LDs in SL-176-treated cells on days 5–8 was 2.33 µm (Figure 2 and Table 1). In mock-treated cells, the proportion of LDs larger than 8 µm and 6–8 µm was 1.9% and 3.1%, respectively. In contrast, in cells treated with SL-176 on days 5–8, the proportion of cells with LD diameters over 8 µm and 6–8 µm was 0.6% and 0.6%, respectively. Thus, PPM1D inhibition in the late stage of differentiation did not affect the amount of LD, but it significantly decreased the LD size. In contrast, SL-176 treatment in the early stage of differentiation strongly decreased both the amount of LD and LD size. These results suggest that PPM1D inhibition affected LD formation via different pathways in the early and late stages of differentiation.

The expression levels of adipogenic transcription factors under each inhibition condition were analyzed using quantitative reverse transcription PCR (RT-qPCR) (Figure 3). In cells treated with the PPM1D inhibitor SL-176 in the early stage of differentiation (days 0–3), the mRNA levels of PPARγ and C/EBPα were lower than those in the cells treated with SL-176 in the middle (days 3–6) and late (days 5–8) stages of differentiation. Thus, PPM1D inhibition in the early stage of differentiation was the most effective in decreasing expression of adipogenic transcription factors.

In the cells in which PPM1D was inhibited in the early and middle stages of differentiation (days 0–5), the mRNA expression level of the cells treated with SL-176 from day 0 to day 5 decreased, while it increased after halting SL-176 treatment on day 5. In the case of PPM1D inhibition in the middle stage of differentiation (days 3–6), the mRNA expression of PPARγ on day 5 were lower than that of the mock-treated cells, the mRNA expression on day 8 was almost the same as that in the mock-treated cells. Thus, PPM1D inhibition in the middle stage of differentiation had a mild effect on the induction of adipogenic transcription factors. The decrease in C/EBPα mRNA expression via the inhibition of PPM1D in the middle stage of differentiation suggests that PPM1D inhibition in the middle stage affects the transcriptional activity of upstream transcription factors of C/EBPα. The cells in which PPM1D was inhibited during the late stage of differentiation (days 5–8), mRNA expression levels of PPARγ and C/EBPα on day 8 did not decrease compared with the mock-treated cells. This result indicates that PPM1D inhibition in the late stage affected neither the expression of adipogenic transcription factors nor adipocyte differentiation. Thus, PPM1D inhibition in the early stage of differentiation strongly decreased the induction of mRNA expression of adipogenic transcription factors PPARγ and C/EBPα contrast, PPM1D inhibition in the middle and late stages of differentiation had little effect on adipocyte differentiation.

On comparing the results of PPM1D inhibition in the early stage (days 0–3) with the those of PPM1D inhibition in the middle stage (days 3–6), it was noted that the amount and the size of LDs decreased to the same extent in both conditions. In contrast, the mRNA expression levels of the adipocyte transcription factors were lower under the condition of PPM1D inhibition in the early stage of differentiation compared with the inhibition in the middle stage of differentiation. These results suggest that the effects of PPM1D inhibition in the early and middle stages of differentiation suppress the formation of LDs via different pathways: one is via the adipocyte differentiation-dependent pathway, and the other is via the adipocyte differentiation-independent pathway.

### 2.2. PPM1D Inhibition in Mature Adipocytes Significantly Reduced the Size of LDs

Next, we analyzed the effects of PPM1D inhibition on mature white adipocytes to understand the effect of PPM1D on adipocyte differentiation-independent LD formation. After 7 days of incubation with the PPM1D inhibitor SL-176 in mature 3T3-L1 white adipocytes, the size of the LDs was dramatically reduced (Figure 4, Appendix A, and Table 2). In mock-treated cells, the fusion of LDs increased the size of the LD from 2.60 µm to 5.47 µm, on average, compared to the size before incubation (Figure 4A,B). In contrast, PPM1D inhibition in mature 3T3-L1 adipocytes decreased the size of the LDs to 2.50 µm, on average. In mock-treated cells, the proportion of LDs having a size over 8 µm and 6–8 µm was 22.5% and 13.1%, respectively (Figure 4B and Table 2). In contrast, PPM1D-inhibited cells had few LDs with sizes of over 8 µm and 6–8 µm; the cells had small LDs with a diameter of 0–2 µm. In the control, the proportion of LDs with 0–2 µm and 2–4 µm size in mock-treated cells was 19.4% and 27.2%, respectively, while in SL-176-treated cells it was 53.7% and 33.1%, respectively. Before incubation with SL-176, mature 3T3-L1 adipocytes on day 8 of differentiation were noted to have a proportion of 46.6% and 37.7% for LDs having the size 0–2 µm and 2–4, respectively (Table 1). This result indicated that treatment with SL-176 in mature 3T3-L1 adipocytes decreased the size of LDs, whereas mock-treated cells exhibited an increase in the size of LDs after 7 days of incubation. On the other hand, the amount of lipids measured by Oil Red O staining was not changed by PPM1D inhibition in mature 3T3-L1 adipocytes (Figure 4C). This result indicated that SL-176 treatment not only suppressed the fusion of LDs but also decreased the size of LDs, whereas it did not affect lipid accumulation.

### 2.3. Small LDs Produced upon PPM1D Inhibition Showed Resistance to Lipolysis

PPM1D inhibition in mature 3T3-L1 adipocytes induced a decrease in the size of the LDs. Several studies have reported that the size of LDs is associated with sensitivity to lipolysis. To examine the difference in sensitivity to lipolysis between small LDs in SL-176-treated cells and large LDs in mock-treated cells, we performed a lipolysis assay in SL-176-treated white adipocytes and mock adipocytes. SL-176-treated mature 3T3-L1 adipocytes were stimulated for lipolysis, and the amount of free glycerol and non-esterified fatty acids (NEFAs) was analyzed (Figure 5). Stimulation by isoprenaline induced lipolysis both in mock-treated cells and in SL-176-treated cells within 2 h after the addition of a lipolysis medium containing isoprenaline (Figure 5A). There were no significant effects in the amount free glycerol and NEFAs released in the medium in response to lipolysis stimulation, although release of NEFAs was slightly decreased in SL-176-treated mature 3T3-L1 adipocytes at 4 h (Figure 5B). This result suggested that PPM1D inhibitor SL-176-treated mature 3T3-L1 adipocytes had small LDs but showed little effects on lipolysis.

### 2.4. LD Size was Regulated by Dephosphorylation of Ser511 of Perilipin 1

The adipocyte-specific LD-associated protein perilipin 1, which is localized on the surface of LDs, regulates LD formation and degradation. To investigate the relationship between PPM1D and perilipin 1, we analyzed the amount and phosphorylation of perilipin 1 in PPM1D inhibitor-treated cells. Mature 3T3-L1 adipocytes were treated with the PPM1D inhibitor SL-176 for 7 days and the mRNA expression of perilipin 1 was quantified via RT-qPCR (Figure 6A). The mRNA expression of perilipin 1 did not change following SL-176 treatment in mature 3T3-L1 adipocytes. The protein expression of perilipin 1 slightly increased by SL-176 treatment in mature 3T3-L1 adipocytes (Appendix A). It was reported that the size of LD and adipocyte lipolysis is controlled by phosphorylation status of perilipin 1 [16]. This suggested that PPM1D inhibition in mature adipocytes reduced the size of LDs by regulating protein function associated with LD formation via dephosphorylation, rather than by reducing the transcription of LD-associated proteins. To understand the mechanism by which PPM1D regulates LD formation, we analyzed the effects of PPM1D inhibition on the phosphorylation of perilipin 1. Ser511 residues of perilipin 1 have been reported to be phosphorylated, as per phosphoproteomic analysis. An in vitro phosphatase assay using phospho-peptide derived from perilipin 1 Ser511 was performed, and it was found that a His-tagged mouse PPM1D(1-413) [His-PPM1D(1-413)] dephosphorylated phospho-Ser511 containing peptide (Figure 6B). In contrast, His-PPM1D(1-413) did not show dephosphorylating activity against phosphorylated peptides containing phospho-Ser492 and Ser517 derived from perilipin 1. SL-176 showed an inhibitory activity against His-PPM1D(1-413) and a phosphorylated Ser511 peptide as a substrate (Appendix A). This result suggests that Ser511 of perilipin 1 is a dephosphorylation target of PPM1D.

To understand the role of phosphorylation of Ser511 of perilipin 1 in LD formation, the phosphorylation mimic mutant S511D and the non-phosphorylation mimic mutant S511A were overexpressed in 3T3-L1 cells, and the formation of LDs was analyzed (Figure 6C,D and Table 3). In the cells that overexpressed S511A, large LDs were formed. In contrast, cells overexpressing S511D contained significantly smaller LDs. The cells overexpressing S511A contained large LDs of over 6 µm in size (2.6%) and 4–6 µm in size (10.2%). In contrast, S511D-expressing cells did not contain large LDs; the proportion of LDs with a size of 4–6 µm and >6 µm was 5.7% and 0.2%, respectively. These results suggest that PPM1D controls LD size via dephosphorylation of the Ser511 residue of perilipin 1.

## 3. Discussion

In this study, we demonstrated that PPM1D controls LD formation via two different pathways. In one, PPM1D regulates adipocyte differentiation process. On the other, PPM1D also regulates function of perilipin 1 in mature adipocytes, resulting of LD formation. We showed that PPM1D controls LD formation via dephosphorylation of perilipin 1 in the adipocyte-differentiation-independent pathway. We noted that inhibition of PPM1D in mature white adipocytes significantly reduced the size of the LDs, but it did not change the total amount of lipids in cells. In this study, it was revealed for the first time that PPM1D dephosphorylates Ser511 of perilipin 1. The C-terminal region including Ser511 of perilipin 1 is a unique region among the perilipin family. Our data revealed that phosphorylation of Ser511 of perilipin 1 affected the size of LDs. Phosphoproteomics analysis has suggested that perilipin 1 Ser511 can be phosphorylated [23], but no phosphatase or kinase that dephosphorylates Ser511 of perilipin 1 has been reported. Moreover, there is no report on the function of phosphorylation at Ser511 of perilipin 1. Our study is the first to report that PPM1D is a specific phosphatase for perilipin 1 Ser511.

PKA is a kinase that targets perilipin 1 [16]. Phosphorylation of Ser517 of perilipin 1 was essential for adipose triglyceride lipase (ATGL)-dependent lipolysis [24]. Ser517 of perilipin 1 was reported to be phosphorylated by PKA, and this phosphorylation inhibited the interaction between perilipin 1 and 1-acylglycerol-3-phosphate O-acyltransferase (also known as CGI-58 or AB-hydrolase-containing 5 (ABHD5)), leading to the activation of adipose triglyceride lipase (ATGL) via interaction with CGI-58 [15]. Ser492 of perilipin 1 was also phosphorylated by PKA and the phosphorylation induces the fragmentation and dispersion of LDs [25]. Therefore, it is suggested that the increase in negative charge at the C-terminus of perilipin 1 by phosphorylation suppresses the hypertrophy of LDs. Phosphorylation state of three Ser residues Ser81, Ser222, and Ser276 of perilipin 1 were also important for forskolin-stimulated lipolysis [26]. It has also been reported that PKA-dependent phosphorylation of six Ser residues (Ser88, Ser222, Ser276, Ser433, Ser492, and Ser517) control hormone-sensitive lipase (HSL)-dependent lipolysis in response to catecholamines in adipocytes [24]. However, only PP1 has been reported to be a phosphatase involved in perilipin 1 dephosphorylation, but the site of dephosphorylation has not been identified [15,27].

In this study, we showed that PPM1D does not target Ser492 and Ser517 but Ser511 of perilipin 1, a novel dephosphorylation site of perilipin 1, and regulates LD formation. The fact that PPM1D targets perilipin 1 and that the target site Ser511 of perilipin 1 was identified, is a notable advancement in elucidating the mechanism underlying LD formation. Analysis using macrophages derived from PPM1D-deficient mice has indicated that PPM1D suppresses the formation of LDs and results in foam cell formation via autophagy-dependent regulation of cholesterol efflux [19]. In cells other than adipocytes, they are also involved in LD formation via a pathway independent from adipocytes, suggesting that PPM1D is an important regulator of LD formation and lipid metabolism in cells.

Inhibition of PPM1D in mature adipocytes resulted in a decrease in LD size, while there was no change in the total amount of lipids. This result suggests that PPM1D is not involved in lipid synthesis and storage but is involved in the fission (fragmentation) and fusion of LDs. The fusion of LDs is a mechanism by which two LDs of different sizes come into contact with each other and fuse to form one large LD; this is one of the mechanisms employed for enlargement of the LD. It has been reported that perilipin 1 promotes the fusion of LDs via Fsp27 during the fusion of LDs. Knockdown of FSP27/Cidea and perilipin 1 has been shown to reduce the size of LDs [14]. It has also been reported that free glycerol released by basal lipolysis and stimulated lipolysis (by epinephrine and theophylline) is increased by FSP27 and perilipin 1 knockdown [28]. FSP27/Cidea overexpression has been reported to increase LD size and total neutral lipid amount. FSP27 controls LD size and determines lipolysis efficiency. If changes in the phosphorylation levels of perilipin 1 alter the protein that interacts with perilipin 1, PPM1D inhibition increases the phosphorylation level of perilipin 1 and changes its interaction with proteins containing Fsp27, leading to suppression of LD fusion. Adipocytes with small LDs induced by PPM1D inhibition showed almost no effects on lipolysis, even though size of LDs was dramatically decreased. It has been reported that small LDs are more susceptible to lipolysis than large LDs [6]. In contrast, it was also reported that lipolysis targets relatively large LDs compared to small LDs and leads to formation of smaller LDs [5]. Adipocytes with small LDs induced through suppression of dephosphorylation of Ser511 of perilipin 1 by PPM1D inhibition. These results suggest that the sensitivity of an LD to lipolysis is not determined by its size. Further studies are required to elucidate the mechanism underlying the lipolysis sensitivity, including effects of PPM1D inhibition on other LD proteins.

In adipocyte-differentiation-dependent pathways, PPM1D regulates adipocyte differentiation in the early stages of differentiation. During adipocyte differentiation, PPM1D inhibition dramatically decreased the amount and size of the LDs. PPM1D inhibition during adipocyte differentiation also downregulated the expression of mRNA of perilipin 1, an adipocyte marker, indicating that PPM1D inhibition suppressed adipocyte differentiation. In the differentiation of preadipocytes to mature adipocytes, adipogenic transcription factors are transcriptionally induced and transcribe the downstream adipogenic marker genes. Transcription factors C/EBPβ and C/EBPδ are first induced in response to differentiation signals, and then, C/EBPα and PPARγ are induced. It has been reported that PPM1D regulates PPARγ function via dephosphorylation [20]. PPM1D inhibition downregulated the expression of PPARγ and C/EBPα, and PPM1D inhibition in the early stages of differentiation showed the strongest suppression of adipocyte differentiation. PPM1D inhibition did not affect the expression of C/EBPβ, which is one of the upstream transcription factors of PPARγ. These results suggest that PPM1D regulates the upstream transcriptional factors of PPARγ, such as C/EBPβ and C/EBPγ. Further insights, such as identification of dephosphorylating substrates for PPM1D in the process of adipocyte differentiation, are needed to elucidate underlying mechanisms.

There are many reports of compounds that affect adipocyte differentiation through the induction of apoptosis in pre-adipocytes. However, few compounds have been shown to affect LD formation in mature adipocytes. We have demonstrated that the PPM1D inhibitor SL-176 regulates LD formation in mature adipocytes via phosphorylation of perilipin 1. SL-176 significantly reduces the size of LDs and affects lipolysis. We also showed that SL-176 significantly inhibited the adipocyte differentiation of pre-adipocytes. These data provide insights into PPM1D inhibitors as lead compounds for anti-obesity therapy via dual pathways.

## 4. Materials and Methods

### 4.1. Cell lines and Materials

3T3-L1 mouse fibroblasts were obtained from the Japanese Collection of Research Bioresource (JCRB, Tokyo, Japan). Plasmids containing the Venus-NLS-2A-perilipin 1(S511A/D) vectors were used for transient expression of perilipin 1 Ser511A and Ser511D mutants. The plasmids contained Venus-NLS and perilipin 1 (S511A) or perilipin 1 (S511D) linked by a 2A self-processing peptide. MDH (AutoDOT) was obtained from abepta (USA). TAP-2CY4E1 was synthesized as previously described [29].

### 4.2. Cell Manipulation

3T3-L1 cells were grown in Dulbecco’s modified Eagle’s medium (DMEM) supplemented with 10% newborn calf serum at 37 °C in an atmosphere of 5% CO_2_. 3T3-L1 cells were seeded at a density of 8 × 10^4^ cells per 35-mm dish. Two days after 100% cell confluence (day 0), the cells were cultured in differentiation medium (MDI) [DMEM containing 10% *v*/*v* fetal bovine serum (FBS) with 100 units/mL penicillin and 100 μg/mL streptomycin (P/S) (Thermo Fisher Scientific, Waltham, MA, USA) with 1.0 μg/mL insulin (Sigma-Aldrich, St. Louis, MO, USA), 1.0 μM dexamethasone (Wako Pure Chemical Industries, Osaka, Japan), and 0.5 mM 3-isobutyl-1-methylxanthine (IBMX, Sigma-Aldrich, St. Louis, MO, USA)]. Cells were placed in adipocyte maintenance medium (DMEM containing 10% FBS with P/S and 1.0 μg/mL insulin) on day 1. Media were exchanged to adipocyte maintenance medium every 2 days until day 8 of induction of differentiation. For experiments with the PPM1D inhibitor (SL-176), the cells were treated with 15 µM SL-176 for the indicated time.

For the experiments involving overexpression of perilipin 1, 3T3-L1 cells were seeded at a density of 8 × 10^4^ cells per 35-mm dish and induced to differentiate. On day 3 of differentiation, 3T3-L1 cells were resuspended in 150 µL of Opti-MEM (Thermo Fisher Scientific, Waltham, MA, USA) containing 20 µg of perilipin 1 expression plasmids. The cells were then electroporated at 500 µF/160 V using the Bio-Rad Gene Pulser Xcell System (Bio-Rad, USA). After electroporation, the cells were seeded on 35-mm dishes in DMEM supplemented with 10% FBS and 1.0 µg/mL insulin. After 24 h, the cells were washed with PBS and cultured in a adipocyte maintenance medium for 2 days. The cells were then fixed, and the size of the LDs was analyzed.

### 4.3. Oil Red O Staining

LDs in the cells were stained and quantified using Oil Red O staining, as previously described [22]. The Oil Red O solution constituted 0.3% Oil Red O/2-propanol:H_2_O in the ratio 6:4 and was filtered through a 0.45 μm PVDF filter (Millipore, Burlington, MA, USA). Then, 5% *v*/*v* 60% isopropanol/H_2_O was added to the filtered solution. Cells were washed once with PBS, followed by washing with 60% 2-propanol and then added to an Oil Red O solution for 30 min. After staining, the cells were washed thrice with 60% 2-propanol/H_2_O. After drying the dishes overnight, extraction of Oil Red O was performed with 100% 2-propanol, once for 60 min and twice for 10 min. For quantification, the absorbance of Oil Red O extract was measured at 490 nm using a microplate reader.

### 4.4. LD imaging and Quantification

3T3-L1 cells were seeded at a density of 1.6 × 10^4^ cells on micro cover glass in a 24-well plate and cultured as described above. On day 8 of differentiation, the cells were fixed with 10% neutral-buffered formalin for 1 h, washed in PBS, and permeabilized with 0.2% Triton X-100/PBS for 15 min. Cells were then incubated with 10 µM MDH for 30 min and then observed via fluorescence microscopy (BZ-9000, KEYENCE, Osaka, Japan) using a DAPI filter (Ex360/40 Dm400 Em460/50). For TAP compound staining, the 3T3-L1 cells were seeded at a density of 2.0 × 10^5^ cells in a 35-mm high µ-Dish (ibidi GmbH, Gräfelfing, Germany) and cultured as described above. The cells were washed with PBS, and then, a 10 µM TAP-2CY4E1 probe was added to Opti-Mem (Thermo Fisher Scientific, Waltham, MA, USA) and incubated at 37 °C for 15 min, followed by observation via fluorescence microscopy (BZ-9000, KEYENCE, Osaka, Japan) using a TRITIC filter (Ex540/25 Dm565 Em605/55) and a GFP filter (Ex470/40 Dm495 Em535/50).

The diameter of the LDs was measured in nine images of MDH-stained cells (approximately 100–200 cells). The LDs in the images were circled and painted black using PowerPoint. Figures that overlapped with each other were removed. The images obtained were used as the binarized images. The size of the circles was measured using Image J. For the experiments involving overexpression of perilipin 1, the cells that showed Venus fluorescence were analyzed.

### 4.5. RT-qPCR

Induction of 3T3-L1 differentiation was performed as previously described. Cells were washed with PBS and lysed with TRIzol reagent (Thermo Fisher Scientific, Waltham, MA, USA). Total RNA was purified according to the manufacturer’s instructions. Total RNA was then reverse transcribed using the PrimerScript II 1st strand cDNA Synthesis Kit (TaKaRa, Kusatsu, Japan) with random hexamer primers according to the manufacturer’s protocols. Quantitative PCR was performed with a CFX96 Touch Real-Time PCR Detection System (Bio-Rad, Hercules, CA USA) using Power SYBR Green Master Mix (Thermo Fisher Scientific, Waltham, MA, USA). The primers used for quantitative PCR were as follows: perilipin 1 forward, GGGACCTGTGAGTGCTTCC; perilipin 1 reverse, GTATTGAAGAGCCGGGATCTTTT; PPARγ forward, CCATTCTGGCCCACCAAC; PPARγ reverse, AATGCGAGTGGTCTTCCATCA; C/EBPα forward, CAAGAACAGCAACGAGTACCG; C/EBPα reverse, GTCACTGGTCAACTCCAGCAC; actin forward, GGCTGTATTCCCCTCCATCG; actin reverse, CCAGTTGGTAACAATGCCATGT.

### 4.6. Protein Purification

His-PPM1D(1-413) was expressed in E. coli BL21 (DE3) pLysS cells and purified by affinity purification and gel filtration, as previously described [21]. The cell pellets were lysed using a French press and then TALON resins (Clontech, Mountain View, CA, USA) with an elution buffer (25 mM HEPES-NaOH pH 6.8, 150 mM imidazole, 200 mM NaCl, 1 mM MgCl_2_, 10% glycerol, and 0.005% Triton X-100) and were used to purify His-PPM1D(1-413). His-PPM1D(1-413) was further purified using a HiTrap Q FF (GE Healthcare, Chicago, IL, USA) column and eluted with IEX start buffer and IEX elution buffer, followed by Superdex 75 (GE Healthcare Bioscience, Chicago, IL, USA) column with SEC elution buffer (25 mM HEPES-NaOH pH 6.8, 500 mM NaCl, 1 mM MgCl_2_, 10% glycerol, and 0.005% TritonX-100).

### 4.7. In Vitro Phosphatase Assay

The phosphatase activity of PPM1D was assayed by measuring the absorbance of the released p-nitrophenol or free phosphate by BIOMOL GREEN (BIOMOL, Hamburg, Germany), as described previously [21]. Briefly, all assays were carried out using 50 mM Tris-HCl pH 7.5, 50 mM NaCl, 0.1 mM EGTA, 0.02% 2-mercaptoethanol, 30 mM MgCl_2_, 40 µM substrate peptide with His-PPM1D(1-413) (2, 5, and 10 nM) for 25 min at 30 °C. The number of phosphatases released was calculated using a standard phosphatase curve. The phosphopeptides perilipin 1(487–498, 492pS) [Ac-WGPARRVS(P)DSFFRP-NH_2_], perilipin 1(506-517, 511pS) [Ac-WGGRAQYS(P)QLRKKS-OH], and perilipin 1(506-517, 517pS) [Ac-WGGRAQYSQLRKKS(P)-OH] were synthesized using Fmoc-standard chemistry and purified [30].

### 4.8. Lipolysis Assay

3T3-L1 cells were seeded at a density of 2.0 × 10^5^ cells in a 35-mm high µ-Dish (ibidi GmbH, Germany), or 4.0 × 10^4^ cells in 24-well plates, and cultured as described above. After differentiation for 8 days followed by treatment with SL-176 for 7 days, the cells were treated with Lipolysis medium [DMEM, 2% fatty acid (FA)-free BSA, 10 µM isoprenaline hydrochloride (Sigma)]. The cells were observed using BZ-9000 (Keyence, Osaka, Japan) at the indicated time points.

For the quantification of free glycerol and NEFAs, after 1 h of incubation with lipolysis medium, the cells were incubated in measurement medium (DMEM, 2% FA-free BSA), and the amount of free glycerol and NEFAs released in the supernatant at 0.5, 1, 2, and 4 h was analyzed using free glycerol reagent (Sigma-Aldrich, St. Louis, MO, USA) and LabAssay NEFA kit (Wako Pure Chemical Industries, Osaka, Japan), respectively.

### 4.9. Western Blotting

Western blotting analysis were carried out as previously described [22]. Briefly, 3T3-L1 cells after differentiation and treatment of SL-176 for 7 days were harvested and cell lysates were separated by SDS-PAGE and transferred to polyvinylidene difluoride membranes. Proteins were detected by enhanced chemiluminescence with the following antibodies: rabbit monoclonal anti-perilipin 1 (D1D8, Cell Signaling Technology, Danvers, MA, USA) and mouse monoclonal anti-actin (C4, Millipore). Secondary antibodies were: anti-mouse IgG HRP-linked antibody (GE healthcare) and anti-rabbit IgG HRP-linked antibody (New England Biolabs, Beverly, MA, USA).

### 4.10. Statistical Analysis

Statistical analysis was performed by one-way analysis-of-variance (ANOVA) followed by Tukey’s post-hoc test, Wilcoxon Rank Sum test, or Student’s *t*-test using R version 4.2.1 (https://www.r-project.org/ (accessed on 23 June 2022)) [31]. For all comparisons, a *p* value of <0.05 was considered statistically significant. * *p* < 0.05, ** *p*< 0.01, *** *p* < 0.001.

## Figures and Tables

**Figure 1 ijms-23-12046-f001:**
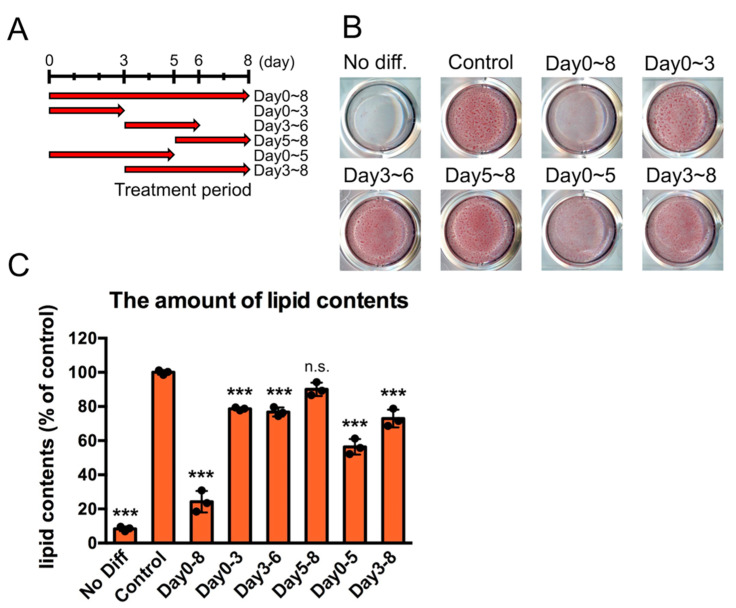
Inhibition of the early stage of adipocyte differentiation dramatically reduced lipid droplet formation and adipocyte differentiation. (**A**) experimental design. The PPM1D inhibitor was added at different stages of adipocyte differentiation. (**B**) Representative Oil red O-stained phase-contrast images of the time-dependent inhibition effect on adipocyte formation. 3T3-L1 cells were induced to differentiation upon stimulation by MDI, with the addition of SL-176 on the indicated days. The cells were fixed and stained with Oil red O. (**C**) Absorbance of Oil Red O extract was measured at 490 nm. Data are presented as the mean ± S.D. values and were obtained from three independent samples in each condition. Significance was analyzed by using one-way ANOVA with a post hoc Tukey’s multiple-comparison test. *** *p* < 0.001, n.s., not statistically significant compared with the control group.

**Figure 2 ijms-23-12046-f002:**
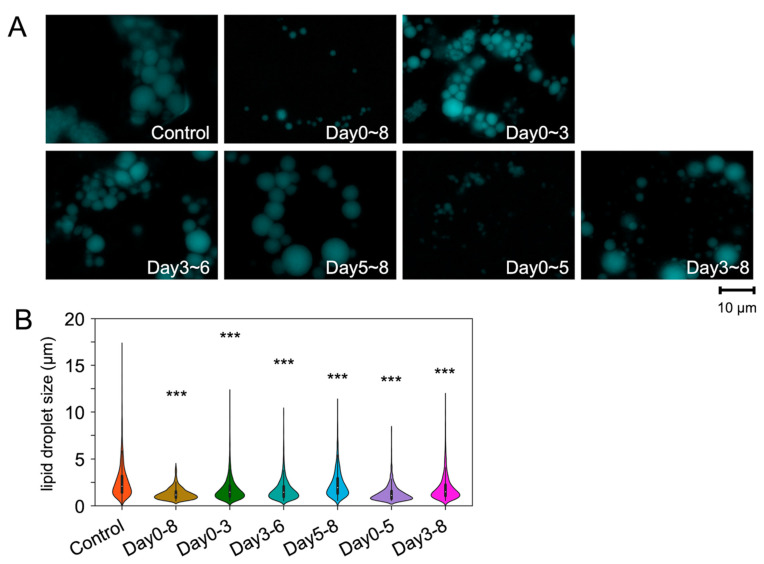
The size of lipid droplets decreased due to PPM1D inhibition in the early stage of differentiation. (**A**) Representative monodansylpentane-stained images of the time-dependent inhibition effect on lipid droplet formation. 3T3-L1 cells were induced to differentiation upon stimulation by MDI, with the addition of SL-176 on the indicated days. (**B**) Size distribution of lipid droplets of 3T3-L1 cells. Violin plot of size distribution of lipid droplets of 3T3-L1 adipocytes. Data were obtained from three independent experiments of 3T3-L1 cells treated with DMSO vehicle (control) or SL-176 (15 µM) for the indicated periods. Significance was analyzed by using Wilcoxon Rank Sum Test. *** *p* < 0.001, compared with the control group.

**Figure 3 ijms-23-12046-f003:**
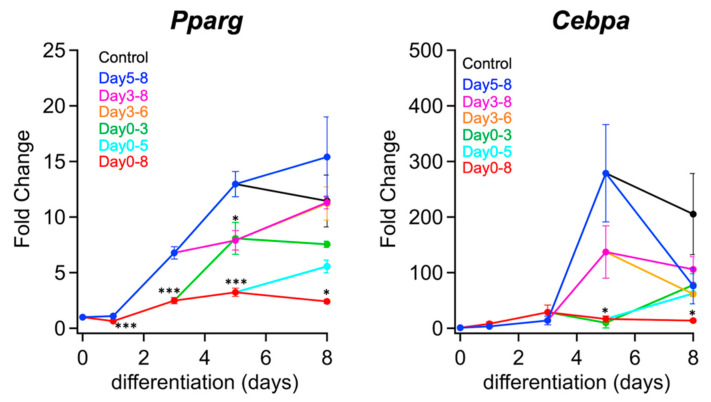
Inhibition of PPM1D in the early stage of differentiation affected by the expression of adipogenic transcription factors. The mRNA expression levels of *Pparg* and *Cebpa* under each condition. The mRNA expression of the indicated genes was measured using RT-qPCR. Black line, without SL-176 (Control); red line, with SL-176 during days 0–8; green line, with SL-176 during days 0–3; cyan line, with SL-176 during days 0–5; orange line, with SL-176 during days 3–6; magenta line, with SL-176 during days 3–8; blue line, with SL-176 during days 5–8. The data was normalized by actin mRNA expression and expressed as fold change. Values are presented as the mean ± S.E.M. of three independent experiments. Significance was analyzed by using one-way ANOVA with a post hoc Tukey’s multiple-comparison test. * *p* < 0.05, *** *p* < 0.001.

**Figure 4 ijms-23-12046-f004:**
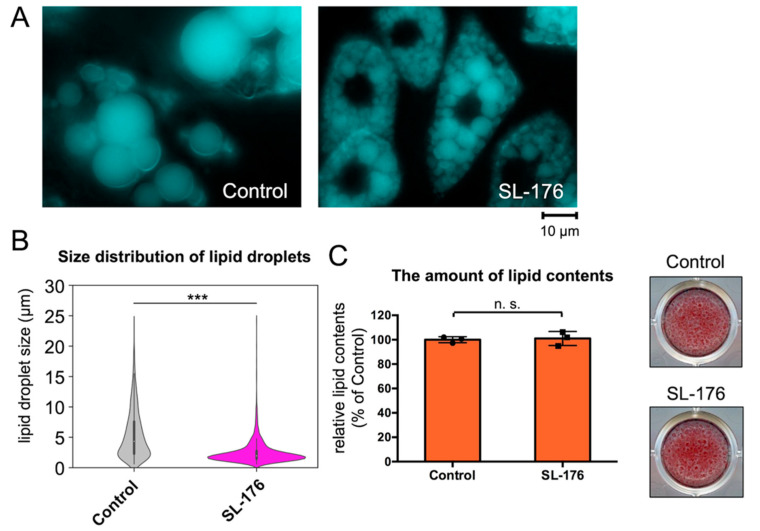
PPM1D-inhibition in mature 3T3-L1 adipocytes decreased the size of lipid droplet. (**A**) Lipid droplet imaging after inhibition of PPM1D. After differentiation for 8 days followed by treatment with SL-176 for 7 days, the cells were stained with monodansylpentane. (**B**) Violin plot of size distribution of lipid droplets of 3T3-L1 adipocytes. Data were obtained from three independent experiments of 3T3-L1 cells treated with DMSO vehicle (control) or 15 µM of SL-176 for 7 days. Significance was analyzed by using Wilcoxon Rank Sum Test. *** *p* < 0.001. (**C**) The lipid content and Oil red O-stained phase-contrast images of 3T3-L1 adipocytes. 3T3-L1 adipocytes were fixed and stained with Oil Red O. Absorbance of Oil Red O extract was measured at 490 nm. Data are presented as the mean ± S.D. of values obtained via three independent samples in each condition; n.s., not statistically significant by Student’s *t*-test.

**Figure 5 ijms-23-12046-f005:**
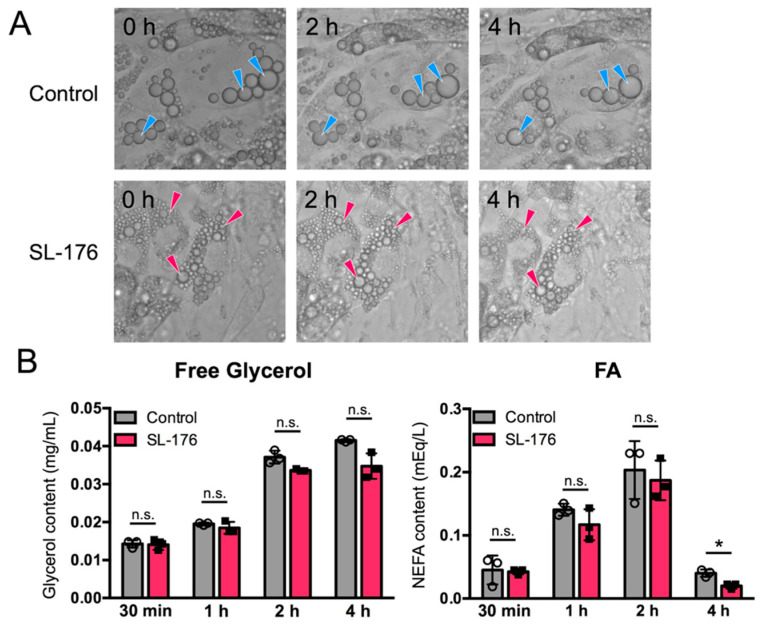
PPM1D-inhibited adipocytes are resistant to lipolysis. (**A**) Cell images during lipolysis assay. The cells were differentiated for 8 days followed by treatment with SL-176 for 7 days; then, the cells were incubated with lipolysis buffer for 2 and 4 h. Arrowheads show the representative LDs that decreased their size. (**B**) The amount of free glycerol and FA were analyzed after 7 days of inhibition of PPM1D in mature 3T3-L1 white adipocytes, then the cells were incubated with lipolysis buffer for 1 h, followed by incubated with maintenance buffer for the indicated time: 30 min and 1, 2, and 4 h. Data represents the mean ± S.D. of three independent experiments. Significance was analyzed by using Student’s *t*-test. * *p* < 0.05; n.s. not statistically significant.

**Figure 6 ijms-23-12046-f006:**
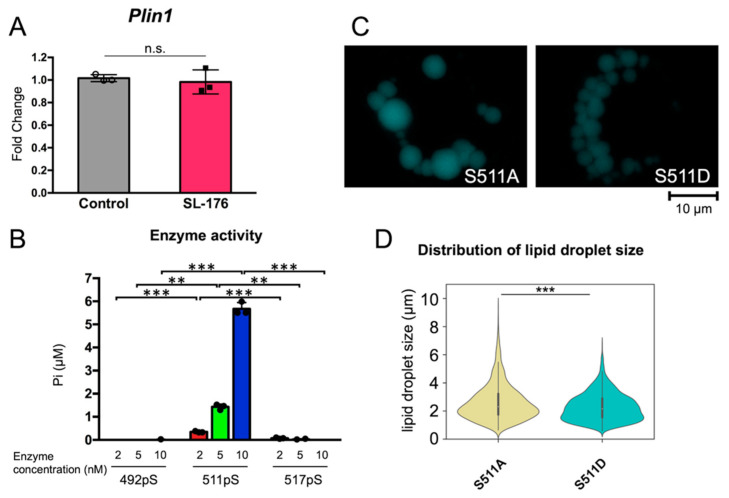
PPM1D regulated lipid droplet formation via dephosphorylation of perilipin 1 at Ser511 residue. (**A**) The expression of white adipocyte marker gene did not change due to the inhibition of PPM1D in mature 3T3-L1 adipocytes. The mRNA expression of perilipin 1 after 7 days of inhibition of PPM1D in mature 3T3-L1 white adipocytes. The mRNA expression level was measured using RT-qPCR. The data was normalized with actin mRNA expression and expressed as fold change. Data represents the mean ± S.D. of three independent experiments. Significance was analyzed by using Student’s *t*-test; n.s., not significance. (**B**) The in vitro enzymatic activity of His-mouse PPM1D against phosphorylated perilipin 1 peptides. Data represents the mean ± S.D. of three independent experiments. Significance was analyzed by using Student *t*-test. ** *p* < 0.01, *** *p* < 0.001. (**C**) Lipid droplet imaging of 3T3-L1 cells that were transfected with perilipin 1 mutants. (**D**) Violin plot of size distribution of lipid droplets of 3T3-L1 adipocytes, overexpressed perilipin 1 mutants. Data were obtained from three independent experiments of 3T3-L1 cells transfected with perilipin 1 S511A or S511D. Significance was analyzed by using Wilcoxon Rank Sum Test. *** *p* < 0.001.

**Table 1 ijms-23-12046-t001:** Average size of lipid droplets (LDs) in each condition.

	Average Size ^1^	Size Distribution of LDs (%) ^2^				
	[µm]	0–2 µm	2–4 µm	4–6 µm	6–8 µm	>8 µm
Control	2.60 ± 0.03	46.6	37.7	10.7	3.1	1.9
Days 0–8	1.30 ± 0.02	88.2	11.4	0.3	0.0	0.0
Days 0–3	1.76 ± 0.02	71.9	23.2	3.7	0.3	0.3
Days 3–6	1.73 ± 0.02	72.7	22.9	3.4	0.2	0.2
Days 5–8	2.33 ± 0.03	52.8	34.7	9.5	0.6	0.6
Days 0–5	1.33 ± 0.02	85.0	13.4	1.5	0.1	0.1
Days 3–8	1.91 ± 0.02	68.3	24.4	5.4	0.5	0.5

^1^ Average size of LD. Data are presented as the mean ± S.E. of values obtained via three independent samples in each condition. Significance was analyzed by using Wilcoxon Rank Sum Test and shown in Figure 2B. ^2^ LD size distribution in 3T3-L1 cells treated with SL-176. LD diameters were calculated using ImageJ software.

**Table 2 ijms-23-12046-t002:** The proportions of lipid droplet (LD) sizes in mature 3T3-L1 cells treated with SL-176.

	Average size ^1^	Size Distribution of LDs (%) ^2^				
	[µm]	0–2 µm	2–4 µm	4–6 µm	6–8 µm	>8 µm
Control	5.47 ± 0.10	19.4	27.2	17.8	13.1	22.5
SL-176	2.50 ± 0.05	53.7	33.1	8.4	2.7	2.1

^1^ Values are presented as the mean ± S.E. of values obtained via three independent samples in each condition. ^2^ Significance was analyzed by using Wilcoxon Rank Sum Test and shown in Figure 4B. LD size distribution in 3T3-L1 cells treated with SL-176. LD diameters were calculated using ImageJ software.

**Table 3 ijms-23-12046-t003:** Size distribution of LDs overexpressing perilipin 1 mutants.

	Average Size ^1^	Size Distribution of LDs (%) ^2^			
	[µm]	0–2 µm	2–4 µm	4–6 µm	>6 µm
S511A	2.61 ± 0.03	38.1	49.0	10.2	2.6
S511D	2.27 ± 0.02	44.5	49.7	5.7	0.2

^1^ Values are presented as the mean ± S.E. of values obtained via three independent samples in each condition. Data were obtained from three independent experiments of 3T3-L1 cells transfected with perilipin 1 S511A or S511D. ^2^ Significance was analyzed by using Wilcoxon Rank Sum Test and shown in Figure 6D. LD size distribution in 3T3-L1 cells overexpressing perilipin 1 S511A or S511D mutants. LD diameters were calculated using ImageJ software.

## Data Availability

All data supporting this study are contained within the article.

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
