# Peer review of "Lipid Droplet Formation Is Regulated by Ser/Thr Phosphatase PPM1D via Dephosphorylation of Perilipin 1"

_ijms, 2022, doi:10.3390/ijms231912046_

Round 1
Reviewer 1 Report
The authors present a manuscript that is relevant for the field. They investigate LD regulation in 3T3-L1 mouse fibroblasts and suggest that lipolysis is not necessarily determined by the size of the LD. They also find a novel role for the phosphorylation of Ser511 in Plin1. This reviewer believes this manuscript presents quality work and compelling work to back their claims.
Major
Lines:
181- Authors state, “SL-176 treatment not only suppressed the fusion of LDs but also decreased the size of LDs, whereas it did not affect lipid accumulation.” Other LD proteins?
208- Authors state, “Compared to mock-treated cells, release of NEFAs in response to isoproterenol stimulation was slightly decreased in SL-176-treated mature 3T3-L1 adipocytes at 4 h (Fig. 5B).” and the wording of “slightly decreased” suggests a change, which is presented preliminarily in their data, though the authors present no statistically significant changes. I would caution against the wording. My recommendation would be to potentially calculate the area under the curve (AUC) during the time measured (30min to 4 h). A statistical difference may be found in that comprehensive analysis and would complement the data presented in individual timepoints.
-Authors present compelling data showing how LD formation is changed after PPM1D inhibition. They present expression levels of Plin1 and the dynamics of LDs. I believe the data presented would benefit from investigating the expression of other perilipin proteins. The phosphorylation data presented here strongly suggests that Plin1ser511 is a pinpointed target of PPM1D, yet the sizing of the LD may be also influenced by the presence of other perilipin proteins. Hence the suggestion for their examination through qPCR or, much preferably, by Western blot, along with Plin1 after 4h of SL-176 treatment.
-Authors suggest in their discussion differential PPM1D targeting of PPARy and C/EBPBeta dephosphorylation and explain how it may affect the transcriptional regulation of their downstream targets. It would be valuable to present data on the phosphorylation sites that are suggested to be affected to complement their mRNA expression data on these transcription factors, as well, they may want to present the gene expression levels of sensitive targets of these transcription factors.
The authors may prefer to present their data for bar graphs as open bars with all datapoints used in their plots presented. For their main findings, this may greatly increase the already transparent policies of the MDPI journal and math that of other publishers.
Minor
Please revise the grammar styling of the manuscript.
Minor comments and examples of the styling issues are presented as follows:
Lines:
12- I believe “dyslipidemia” should be separated from the list of diseases, as I understand that is a condition, not necessarily a disease.
21- change “…formation found…” to “…formation was found…”
43- citation is needed for lines starting at 43 and 44.
50- “regulated in cells” to “regulated in different cell types” if that conveys better the authors interpretation of the literature
83- change “PPM1D protein increases in obesity model mice“ to “PPM1D protein increases in mice models of obesity”
Fig. 1- modify the diagram on panel A to align the day indicators with the arrows or make it clearer that the “dayX~X” corresponds to each arrow.
Table 1. In data from “Day 0-8”, at 6-8um, “0.0.”, remove the extra period
268- Change the starting line of the paragraph to make it clearer “In this study, we demonstrated that PPM1D controls LD formation via two different pathways: PPM1D regulates adipocyte differentiation process. PPM1D also regulates function of perilipin 1 in mature adipocytes, resulting of LD formation.” To “In this study, we demonstrated that PPM1D controls LD formation via two different 268 pathways. In one, PPM1D regulates adipocyte differentiation process. On the other, PPM1D also regulates function of perilipin 1 in mature adipocytes, resulting of LD formation.” Or modify as it better captures the authors interpretation.
Reviewer 2 Report
In the manuscript entitled “Lipid droplet formation is regulated by Ser/Thr phosphatase PPM1D via dephosphorylation of perilipin1”, the author investigates the effects of PPM1D inhibitor SL-176 on lipid droplets formation in 3T3-L1 adipocyte and found that treatment of mature adipocytes with SL-176 can reduce lipid droplets formation without any effect on isoprenaline induced lipolysis. Results from this manuscript are interesting, however, some of the results are preliminary and the conclusions are overstated, which detracts from the overall quality. I have some comments shown below to help improve the clarity of this manuscript.
- The conclusion described in this manuscript” Lipid droplet formation in regulated by Ser/Thr phosphatase PPM1D via dephosphorylation of perilipin 1” is overstated. More experiments are needed to get such a conclusion. Does PPM1D (or SL-176) have any effects on the formation of LDs in mature adipocytes with S511D or S551 perilipin? The specificity of SL-176 as a PPM1D inhibitor still needs more testing, do other PPM1D inhibitors, such as M321237, CCT00793, or GSK283071 have the same effects on lipid droplet formation?
- It would be better to show the P value and statistical significance in Table 1, Table 2, Table 3, Figure 3, and Figure 6B.
- In Figure 3, it would be better to show the protein levels of PPARg and C/EBPa under each condition. Does the perilipin protein level change under each condition?
- For Figure 5A, it would be better to annotate what the arrow presents in the figure legend.
- Does SL-176 have any effect on basal lipolysis?
- For Figure 6A, it would be better to show the protein level of Perilipin.
- The conclusion that PPMD1 regulated LD size via dephosphorylation of Ser511 of Perilipin in Line 221 is overstated. Does the phosphorylation of Ser511 of Perilipin1 change after CL-176 treatment? Does SL-176 have any effects on lipid droplet formation in 3T3-L1 mature adipocytes with S511A perilipin1?
- Do S511A and S511D affect the early differentiation of 3T3-L1 adipocytes? It would be better to use the inducible system, in which S511A or S511D perilipin 1 will be overexpressed in the late stage of differentiation (mature adipocyte).
Round 2
Reviewer 2 Report
The authors have addressed some of my concerns in the revised manuscript. The title and major conclusion in the revised manuscript is that lipid droplet formation is regulated by PPMD1 via dephosphorylation of perilipin1, however, to get such a conclusion, more experiments are still needed as commented before.
1) The author should demonstrate that SL-176 treatment can inhibit the dephosphorylation of Ser511 of perilipin 1 in the mature adipocyte or SL-167 could inhibit the in vitro enzymatic activity of PPM1D against phosphorylated perilipin 1 peptide.
2) Does SL-176 treatment have any effects on the LD formation in 3T3-L1 transfected with S511A or S511D?
Round 3
Reviewer 2 Report
The authors have answered my questions and I have no further comments.